# Program Semantic Inequivalence Game with Large Language Models

## Abstract

Large Language Models (LLMs) can achieve strong performance on everyday coding tasks, but they can fail on complex tasks that require non-trivial reasoning about program semantics. Finding training examples to teach LLMs to solve these tasks can be challenging.

In this work, we explore a method to synthetically generate code reasoning training data based on a **semantic inequivalence game** (**SInQ**): a **generator** agent creates program variants that are semantically distinct, derived from a dataset of real-world programming tasks, while an **evaluator** agent has to identify input examples for which they behave differently. The agents train each other semi-adversarially, improving their ability to understand the underlying logic of code.

We evaluated our approach on multiple code generation and understanding benchmarks, including cross-language **vulnerability detection** (Lu et al., 2021), where our method improves vulnerability detection in C/C++ code despite being trained exclusively on Python code, and the challenging **Python builtin identifier swap** benchmark (Miceli Barone et al., 2023), showing that whereas modern LLMs still struggle with this benchmark, our approach yields substantial improvements.

We release the code needed to replicate the experiments, as well as the generated synthetic data, which can be used to fine-tune LLMs.

## 1 Introduction

Assistants based on Large Language Models (LLMs) are widely used by programmers for coding tasks. While they perform well on common tasks, they still struggle with non-trivial reasoning about program semantics (Miceli Barone et al., 2023; Maveli et al., 2025). This limitation can lead to subtle bugs or prevent the detection of preexisting vulnerabilities and adversarial backdoors (Dinh et al., 2023; Dou et al., 2024), ultimately compromising the safety and security of generated code (Wang et al., 2024; Mohsin et al., 2024).

LLMs' code generation and understanding capabilities are typically improved by fine-tuning on a mixture of human-annotated and synthetically generated data. For example, the Llama-3 recipe (Llama3, 2024) provides a prototypical approach. However, human annotation is expensive and fails to cover many non-trivial scenarios. Typical synthetic generation approaches rely on LLMs to generate coding problem statements along with corresponding solutions and unit tests, then validate solutions by executing them against the tests. While this allows for large-scale dataset creation, it may still provide limited coverage of problem types and introduce noise, as unit tests do not always align well with problem statements, particularly in edge cases.

Self-play involves training AI agents by pitting them against each other in adversarial games, incentivizing them to discover and defend against unusual scenarios. This approach has enabled AI systems to achieve human-level or even superhuman performance in games such as Go (Silver et al., 2016; 2017b), Chess (Silver et al., 2017a), Dota 2 (OpenAI et al., 2019), StarCraft II (Arulkumaran et al., 2019), and even social games involving dialogue, such as Diplomacy (FAIR, 2022). However, self-play methods typically require external engines to enforce game rules and compute scores, making them challenging to apply to open-ended tasks

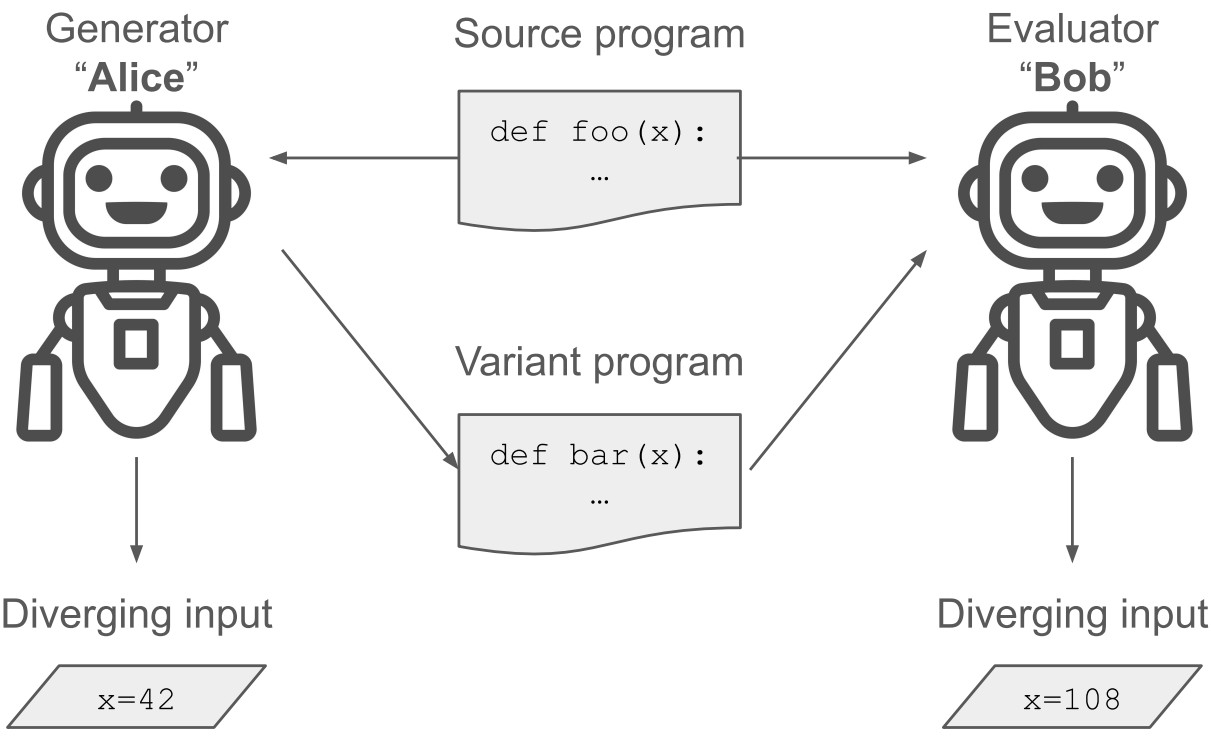

Figure 1: Semantic inequivalence game: Alice receives a source program $P$ and generates a variant program $Q$ and a diverging input. Bob receives $P$ and $Q$ and generates another diverging input.

like coding. Recreational competitive coding environments such as CROBOTS[1] are overly domain-specific and impose strict performance limits, making them unsuitable for training agents in general code reasoning. We are aware of only one work, Zhao et al. (2025), concurrent to our own, which uses self-play to train LLMs for arbitrary code generation, while Dong & Ma (2025) use a similar approach for theorem proving.

In this work, we introduce a game based on program semantic inequivalence designed to train agents in code reasoning across arbitrary domains. By design, this game has no theoretical performance cap. We use it to train LLMs through self-play, demonstrating significant performance improvements on challenging tasks.

## 2 Proposed method

Our approach involves two LLM agents engaging in a game where the **generator** agent, "Alice," creates challenging code understanding problems for the **evaluator** agent, "Bob," to solve. Alice's goal is to deceive Bob into making mistakes, requiring her to generate difficult instances. However, Alice must also provide solutions, meaning the instances cannot be unsolvable or excessively difficult. We train Alice to become more effective at misleading Bob and Bob to become better at resisting deception, encouraging both agents to develop a deeper understanding of program semantics.

Our approach is based on the **semantic equivalence** of programs, or more specifically, **semantic inequivalence**. This allows for precise verification of solutions, unlike problems based on natural language specifications or unit tests, which offer only partial coverage. Moreover, it is fundamentally linked to computability theory through reductions to Rice's theorem and the Halting problem.

In practical applications, reasoning about program equivalence and inequivalence is valuable for identifying bugs and security vulnerabilities introduced during code refactoring.

---

[1]https://github.com/tpoindex/crobots

## 2.1 The semantic inequivalence game

Two programs $P$ and $Q$ are semantically equivalent if, for any given input $x$, either they both halt producing the same output $y$ or neither halts. Determining semantic equivalence is a fundamental problem in program verification and compiler design, but automatic proving equivalence between arbitrary programs is complicated since popular programming languages, such as Java and Python, are defined through natural language specifications or reference implementations rather than formal semantics. Even when formal semantics exist for a subset of a language, automatically generating fully machine-checkable equivalence proofs for non-trivial code is quite challenging even for expert human programmers. We sidestep this issue by defining a program understanding game that focuses solely on inequivalent programs.

We define the **Semantic Inequivalence Game** as the following one-shot interaction between two players: the **generator**, "Alice," and the **evaluator**, "Bob":

1. Alice receives a program $P$ and generates another program $Q$, which has to be inequivalent to $P$, along with a diverging input $x$ such that $P(x) \neq Q(x)$.

2. The diverging input is verified by executing both programs on it. If $P(x) = Q(x)$, Alice loses.

3. Bob receives $P$ and $Q$ and attempts to produce a diverging input $\hat{x}$ (which may or may not be the same as $x$). If Bob correctly identifies a diverging input, he wins and Alice loses; otherwise, Bob loses and Alice wins.

Alice's objective is to generate challenging instances for Bob, while Bob's goal is to solve them. In this game, correctness can be verified simply by executing the programs on the provided diverging inputs, eliminating the need for generating and verifying formal proofs.

Both agents are trained iteratively through repeated play. The source programs $P$ provided to Alice are sampled from a dataset, such as a collection of short, self-contained programming exercises spanning a variety of tasks (e.g., MBPP (Austin et al., 2021)). This ensures that the game stays **grounded** to real-world coding problems. If Alice were allowed to generate both $P$ and $Q$, she might develop an incentive to produce unusual, obfuscated code that might not contribute to Bob's general reasoning skills.

To approximate non-termination detection, we impose a randomized time limit that significantly exceeds the typical runtime of source programs. This prevents Alice from exploiting a fixed time limit, for example, by generating a program $Q$ that loops for a predetermined duration before returning the same output as $P$.[2]

**Example 1.** [3] *1. Alice receives program P:*

```
def fib(n):
    if n <= 0:
        return 0
    elif n == 1:
        return 1
    return fib(n - 1) + fib(n - 2)
```

*and returns program Q:*

```
def fib(n):
    if n == 0:
        return 0
    elif n == 1:
        return 1
    return fib(n - 1) + fib(n - 2)
```

---

[2]The time limit is randomized to discourage Alice from gaming the system by crafting artificial delays, which could lead to uninteresting cases.

[3]This example is artificial, please refer to section 3.3 for an analysis of a real example from the MBPP dataset.

*together with diverging input x:*

*{ "n" : −1}*

*2. Both programs are run in a sandbox with input x, which results in P returning 0 while Q recurs until it triggers either the recursion limit of the Python interpreter or the randomized time limit, proving that x is indeed a diverging input.*

*3. Bob receives both P and Q a generates a possibly different diverging input x̂, e.g.:*

*{ "n" : −2}*

*P and Q are evaluated on input x̂, proving that this is also a diverging input, therefore, Bob wins this round.*

Unlike games such as Go or Chess, where perfect play is theoretically possible, the semantic inequivalence game has no strict performance cap: given an infinite time limit, Bob's task is undecidable (see Appendix A). This implies that in principle the agents can learn arbitrarily complex coding logic while remaining grounded in a dataset of practical, real-world programming problems.

The semantic inequivalence game is entirely adversarial and essentially a zero-sum game, provided that Alice generates only valid outputs. In some cases, modifying the game to be positive-sum may be beneficial, both to facilitate integration with supervised fine-tuning (SFT) and to prevent degenerate strategies (e.g., Alice generating excessively difficult cryptographic puzzles). We discuss these considerations further in Appendix B.

## 2.2 Implementation with Supervised Fine-tuning and Difficulty Targeting

The semantic inequivalence game, as defined above, is well-suited for reinforcement learning, however, reinforcement learning was not available on the OpenAI API at the time of our experiments, therefore, we devised the following rejection sampling fine-tuning implementation, with explicit difficulty supervision.

When we present the program pair $(P, Q)$ to Bob, we can sample $N$ evaluation responses and define the difficulty of the pair based on the number of correct assessments:

$$d(P, Q, Bob) = 10 \left(1 - \frac{N_{\text{correct}}}{N}\right)$$

For example, if Bob can solve the pair $(P, Q)$ 40% of the time, the difficulty of this instance is 6.

During generation, we ask Alice to generate a program with a specific target difficulty $d_t$, usually set to the maximum value of 10 (though in some cases, we may set it to a lower value, making the game positive-sum; see Appendix B).
Let:

$$I = Template_{Alice}(P, d_t)$$
$$O = Alice(I)$$
$$(CoT, Q, x) = Extractor_{Alice}(O)$$

If Alice's output is invalid, we discard it. Otherwise, we evaluate it with Bob to estimate its actual difficulty. We then create a new SFT training example for Alice by substituting the estimated difficulty with the target difficulty in the input. That is, we treat the actual generated program's difficulty as if it were the target difficulty:

$$TrainingExample_{Alice} := (Template_{Alice}(P, d(P, Q, Bob)), O)$$

We can generate one or more training examples for Alice from the $P$ programs in the source program set, then batch-train Alice, for instance, using OpenAI's fine-tuning API with the chat LLM format. Here, the input is encoded as the "user" message and the output as the "assistant" message, with the same "system" message used during generation. The loss is minimized only on the "assistant" message.

We can continue to extend this dataset across multiple generations of Alice, as long as Bob remains unchanged. Once we update Bob (using rejection sampling SFT on its own successful input-output pairs), we need to recompute all the difficulty estimates for the programs in Alice's dataset, as Bob is presumably stronger. We have found it beneficial to train Alice for many epochs before training a new Bob. Initially, Alice tends to generate examples that are too easy for Bob (especially when both Alice and Bob are based on the same LLM). Ideally, we would continue training Alice until convergence before each new Bob training run.

Since Alice's initial examples are often very easy for Bob (with difficulty zero for most), using all of them as training examples would overwhelm Alice's training set with unhelpful instances. This could be detrimental, as we would minimize the loss on tokens of programs that we don't want Alice to generate. To address this, we bias the training set by selecting all hard examples, defined as $d(P, Q, Bob) \geq 5$, i.e., the examples that fool Bob at least 50% of the time, plus a fraction of the remaining examples (20% of the number of hard examples), sampled without replacement by going through difficulty bins in a round-robin fashion.

We also explicitly train Alice to predict instance difficulty by including training examples in the following format:

$$TrainingExample_{diff} := (Template_{Alice}(P, \texttt{"Any"}), O,$$
$$Template_{diff_{in}}, Template_{diff_{out}}(d(P, Q, Bob))))$$

where the target difficulty in the first "user" message is replaced by the string "Any", and the first "assistant" message contains Alice's self-generated output instance. The second "user" message provides an instruction to predict the difficulty of the instance, and the second "assistant" message contains the actual difficulty. We minimize the loss only on the second "assistant" message. This part of the dataset is also biased towards hard examples, but we set the number of easy examples at 50%, as we are not minimizing the loss on the tokens of easy instances but only on their difficulty prediction. Therefore, including these examples is unlikely to be detrimental.

Refer to Appendix C for all the templates used to interact with the LLM.

## 3 Experiments

### 3.1 Training

We run our main set of experiments using OpenAI `gpt-4o-mini-2024-07-18` as the base LLM for both Alice and Bob. In order to train our agents, we use the training portion of MBPP as our source set of programs, using only the `code` field from each source example. We perform additional experiments using OpenAI `gpt-4.1-nano-2025-04-14` as the base LLM.

Both Alice and Bob are instructed to produce output in markdown format, using markdown sections to distinguish their CoT traces from the final outputs, which are embedded in Python code blocks. We sample from the models with a temperature of 1.0 and top_p = 0.7, generating $N = 10$ samples per query. We use the Mistune markdown parser[4] followed by the Python `ast` parser/unparser. This step extracts, syntactically validates, and normalizes the outputs[5]. We then semantically check the diverging inputs against the pairs of source and generated programs by executing them within a test harness, using a randomized time limit, uniformly sampled between 2.5 and 5.5 seconds, to discourage Alice from generating instances that rely on race conditions.

We train the models via SFT with difficulty targeting (always set to 10) and difficulty prediction, as described in Section 2.2, using the default hyperparameters of the OpenAI fine-tuning platform.

For initial set of the experiments on the `gpt-4-mini` models, we perform 7 batched training rounds for Alice, followed by a single training round for Bob. This was due to time and financial constraints; ideally, we would

---

[4] https://mistune.lepture.com/en/latest/
[5] Parsing and then unparsing the Python code with `ast` removes comments or unusual indentation styles, preventing trivial non-semantic attacks.

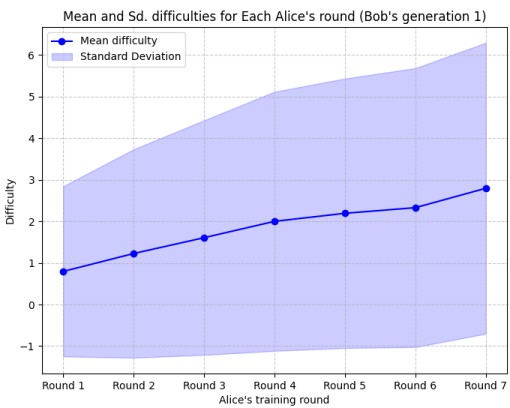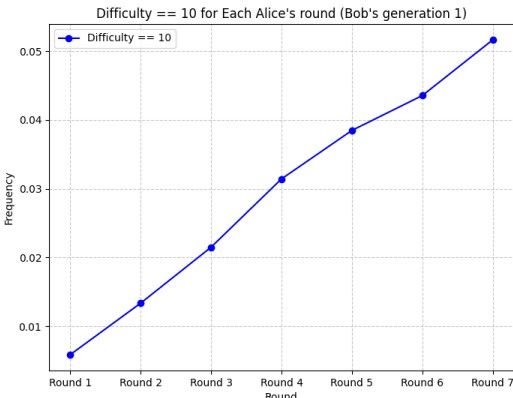

Figure 2: Instance difficulty with respect to an untrained Bob, plotted against the number of Alice's training rounds, `gpt-4-mini` models. Left: mean and standard deviation. Right: fraction of instances with maximum difficulty.

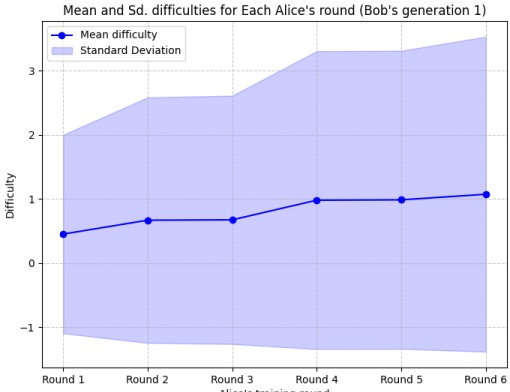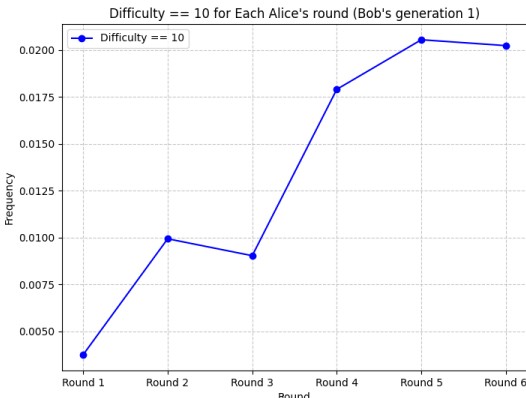

Figure 3: Instance difficulty with respect to an untrained Bob, plotted against the number of Alice's training rounds, `gpt-4.1-nano` models. Left: mean and standard deviation. Right: fraction of instances with maximum difficulty.

perform training rounds for Alice until convergence of the average instance difficulty before performing one training round for Bob. For the additional set of experiments on the `gpt-4.1-nano` models, we perform 6 batched training rounds for Alice, which is sufficient for convergence, followed by a single training round for Bob. We report the difficulty curves in Figure 3.1 and 3.1.

Each of Alice's training runs starts from the baseline LLM checkpoint rather than the previous fine-tuned checkpoint, but we accumulate instances to be used as training examples between rounds. Since Bob does not change between Alice's training rounds, the difficulty estimate of each instance remains valid. However, if we were to perform additional training rounds for Alice after training Bob, we would have to either discard the training set or re-estimate the difficulty of each instance by evaluating it with the new Bob.

We consider the fine-tuned Bob to be our final model, which we use for evaluation.

We also attempted a set of experiments based on the Alibaba Qwen3 Thinking LLM, specifically the smallest non-quantized version available on Hugging Face `Qwen3-4B-Thinking-2507`. We performed generation using the recommended hyper-parameters, with full context length (32768 tokens). We used a rank-stabilized LoRA

adapter with rank = 128, alpha = 16, dropout probability = 0.1, epochs = 5, maximum learning rate = $2e-4$. We did not use difficulty prediction examples, since according to the Qwen3 fine-tuning guidelines, CoT traces should not be included in assistant messages in multi-turn conversations except on the last one. This sets of experiments lead to unstable results, with the proportion of examples with maximum difficulty increasing with each Alice's round, as expected, but the mean difficulty decreasing. Training curves are reported in Figure **??**. This may be due to the small size of the model or the adaptor. Due to financial constraints, we are unable to evaluate larger models.

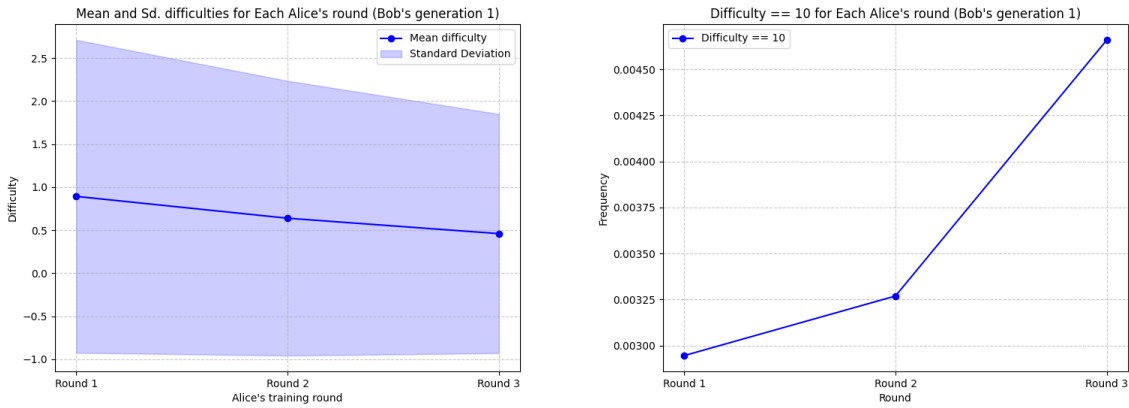

Figure 4: Instance difficulty with respect to an untrained Bob, plotted against the number of Alice's training rounds, `Qwen3-4B-Thinking-2507` models. Left: mean and standard deviation. Right: fraction of instances with maximum difficulty.

## 3.2 Intrinsic Evaluation

Our goal is to improve our model's performance on code understanding tasks. In this section, we report how much better our evaluator model, Bob, performs on the semantic inequivalence game after its first and only training round. We use the final trained generator model Alice (from round 7) to generate the challenge instances. These instances are created using source programs from either the training portion of the MBPP dataset, as done during training, or from the test portion of MBPP, which neither Alice nor Bob have seen before. The results are reported in Table 1.

| Source programs | Untrained | Trained |
|---|---|---|
| MBPP Train | 75.99% | **86.98%** |
| MBPP Test | 88.37% | **91.67%** |

Table 1: Percentages of semantic inequivalence game instances solved by Bob, without or with training, `gpt-4-mini` models.

We observe that while the performance of the untrained Bob (baseline `gpt-4o-mini-2024-07-18`) is already high, this is expected because we did not train Alice to convergence. However, performing a single training round for Bob substantially improves its ability to play the game.

This demonstrates that our training protocol is effective in teaching LLMs to reason about the inputs that cause different variants of a program to behave differently.

## 3.3 Qualitative analysis

We manually analysed the some of Alice's outputs in order to understand what kind of manipulation it introduces. A typical pattern that we find is the introduction of subtle bugs in conditional branches that deal with edge cases. For instance, starting from the following program from MBPP:

```python
from sys import maxsize

def max_sub_array_sum(a, size):
    max_so_far = -maxsize - 1
    max_ending_here = 0
    start = 0
    end = 0
    s = 0
    for i in range(0, size):
        max_ending_here += a[i]
        if max_so_far < max_ending_here:
            max_so_far = max_ending_here
            start = s
            end = i
        if max_ending_here < 0:
            max_ending_here = 0
            s = i+1
    return (end - start + 1)
```

Alice generates:

```python
from sys import maxsize

def max_sub_array_sum_y(a, size):
    max_so_far = -maxsize - 1
    max_ending_here = 0
    start = 0
    end = 0
    s = 0
    for i in range(0, size):
        max_ending_here += a[i]
        if max_so_far < max_ending_here:
            max_so_far = max_ending_here
            start = s
            end = i
        if max_ending_here < 0:
            s = i + 1
    return end - start + 1
```

These programs look superficially the same, they only difference is that the generated variant lacks the `max_ending_here = 0` statement in the `if max_ending_here < 0:` branch inside the loop, which causes it to mishandle negative values in the array `a`.

This sort of bugs often occur in programs that have security vulnerabilities, therefore we believe that by being trained on such examples, Bob can learn to reason on the semantics of vulnerabilities. This could explain the improvements we observe on vulnerability datasets (Section 3.4.2).

### 3.4 Extrinsic Evaluation

Being proficient at playing the semantic inequivalence game may be directly useful in certain circumstances, such as determining whether a code refactoring has introduced subtle bugs. However, ultimately, we aim for this game to teach LLMs skills that generalize to a variety of tasks. Therefore, we evaluate our approach across a range of code-related tasks using standard benchmarks.

While we include code generation tasks, our primary focus is on code understanding. Therefore, we use the trained evaluator Bobs, denoted as `sinq-gpt-4o-mini` (based on `gpt-4o-mini`), and `sinq-gpt-4o-mini`

(based on `gpt-4.1-nano`) as our main checkpoints. Each trained model is primarily compared to its own base model. We do not evaluate the models based on `Qwen3-4B-Thinking-2507`, as that set of experiments yielded inconsistent training curves and was aborted.

### 3.4.1 Python Builtin Identifier Swap

The **Python builtin identifier swap** is a very challenging code understanding benchmark introduced by Miceli Barone et al. (2023). In its classification version, each example consists of two variants of a Python snippet, with an instruction asking the model to determine which variant is more likely to be correct. The challenge is that the snippets are prepended with a statement that reassigns two builtin Python functions used in the code, e.g.

```
print, len = len, print
```

One of the two snippets is a Python function extracted from a GitHub repository, while the other is the same function with all instances of the two builtin identifiers (e.g., `len` and `print`) swapped. Because of the reassignment of the two identifiers, the modified snippet is correct but highly out-of-distribution, while the original snippet is in-distribution but incorrect. Miceli Barone et al. (2023) found that this confused all the state-of-the-art LLMs available at the time, to the point that they even performed worse as their size increased, a case of **inverse scaling** (McKenzie et al., 2023).

We evaluate `gpt-4o-mini` and `gpt-4.1-nano`, which had not been released at the time of the original study, and our own `sinq-gpt-4o-mini` and `sinq-gpt-4.1-nano` (trained Bobs) on this benchmark. We use either the original prompt template or a variant that instructs the models to perform chain-of-thought reasoning before answering. We report our results in Table 2.

| Base LLM | Baseline | **SInQ** | Baseline CoT | **SInQ CoT** |
|---|---|---|---|---|
| gpt-4o-mini | 1.65% | **5.35%** | 1.90% | 2.30% |
| gpt-4.1-nano | 2.80% | 7.40% | 14.35% | **15.30%** |

Table 2: Accuracy results on the Python builtin identifier swap benchmark for the baseline models (untrained Bobs) and our **SiNQ** models (trained Bobs), with or without chain-of-thought.

We observe that, despite `gpt-4o-mini-2024-07-18` and `gpt-4.1-nano-2025-04-14` being released years after this benchmark was published, they still performs very poorly. In fact, they perform than GPT-3.5 (3.35% accuracy)[6], indicating that this benchmark remains challenging. Our approach yields a substantial improvement (+3.7%/+4.6%) over the baseline without using CoT. Surprisingly, the improvement when using CoT is smaller (+0.4%/+0.1).

This benchmark is quite different from the synthetic data used to train our model in the **semantic inequivalence game**. The main similarity is that both tasks involve reasoning about the semantics of unusual snippets of Python code. The substantial improvements we observe indicate that our approach teaches the model generalizable code reasoning skills.

We report additional results on this benchmark with state-of-the-art reasoning models in Appendix D.

### 3.4.2 Vulnerability Detection

Security vulnerabilities in code often arise from counterintuitive behaviours, where the intuitive understanding that programmers, whether human or LLM, have of the code's semantics differs from its actual semantics in edge cases that evade pre-deployment testing. Our semantic inequivalence game incentivises the generator Alice to find edge cases that fool the evaluator Bob, who is then incentivised to become more robust by improving his reasoning about code semantics. Ideally, these capabilities should generalize to security vulnerability detection.

---

[6]Raw results for Miceli Barone et al. (2023) are available on the GitHub repository associated with the paper: `https://github.com/Avmb/inverse_scaling_prize_code_identifier_swap/blob/main/eval_chat_llms/eval_chat_llms_results.json`.

We evaluate our approach by testing it on two vulnerability detection benchmarks.

**PySecDB** (Sun et al., 2023) is a dataset of commits, represented as diff patches, for Python programs, classified as either containing or not containing a security fix. We present these patches to the LLMs, instructing the models to classify them. We do not provide the rest of the repository as a reference, making this a challenging task. Since some of these commits are quite long, we discard those that exceed the maximum context length of 128,000 tokens.

| Base LLM | Baseline | **SInQ** | Baseline Maj | **SInQ Maj** | Baseline CoT | **SInQ CoT** |
|---|---|---|---|---|---|---|
| gpt-4o-mini | 82.43% | 82.51% | 82.48% | **82.81%** | 73.55% | 73.00% |
| gpt-4.1-nano | 83.20% | 83.33% | 83.03% | **83.35%** | 78.74% | 78.20% |

Table 3: Vulnerability detection results on the PySecDB benchmark, with or without majority voting of 9 or chain-of-thought.

**CodeXGLUE Defect Detection** (Lu et al., 2021) is a dataset of code snippets in C/C++ classified according to whether they contain known security vulnerabilities. This is a particularly challenging dataset for our approach, as we fine-tuned our model only with Python code.

| Base LLM | Baseline | **SInQ** | Baseline Maj | **SInQ Maj** | Baseline CoT | **SInQ CoT** |
|---|---|---|---|---|---|---|
| gpt-4o-mini | 55.23% | 55.60% | 55.12% | **56.04%** | 47.69% | 47.22% |
| gpt-4.1-nano | 54.76% | **55.27%** | 54.83% | 55.23% | 52.20% | 49.85% |

Table 4: Vulnerability detection results on the CodeXGLUE Defect Detection benchmark, with or without majority voting of 9 or chain-of-thought.

We run our experiments using standard greedy classification (temperature = 0.0, no CoT), majority voting of 9 (temperature = 1.0, $N = 9$, no CoT), and CoT mode (temperature = 0.6, $N = 1$). The results are reported in Tables 3 and 4.

Our approach yields small but consistent improvements across two datasets, with different tasks and programming languages. These results suggest that our models have acquired additional capabilities in reasoning about security vulnerabilities, despite not having been specifically trained for this task.

### 3.4.3 Code Generation

We run a standard code generation experiment using the EvalPlus harness (Liu et al., 2023; 2024), which evaluates LLMs on the test portions of MBPP and HumanEval (Chen et al., 2021), as well as on augmented versions of these datasets, MBPP+ and HumanEval+, which contain additional unit tests per instance. The results are reported in Table 5.

| | Base LLM | | | |
|---|---|---|---|---|
| Test set | gpt-4o-mini | | gpt-4.1-nano | |
| | Baseline | **SInQ** | Baseline | **SInQ** |
| MBPP | 82.8% | **84.9%** | **87.6%** | 86.0% |
| MBPP+ | 69.6% | **70.4%** | **72.5%** | 72.2% |
| HumanEval | **87.2%** | **87.2%** | 89.6% | **90.2%** |
| HumanEval+ | **82.9%** | 82.3% | 84.1% | **87.2%** |

Table 5: Pass@1 rates on the EvalPlus suite, for the baseline models (untrained Bobs) and our SInQ models (trained Bobs).

We observe that for `gpt-4o-mini` our approach substantially improves code generation Pass@1 accuracy on both the original MBPP test set (+2.1%) and the more challenging MBPP+ version (+0.8%). It maintains the same level of accuracy on HumanEval and loses a slight amount of accuracy on the more difficult

HumanEval+ (-0.4%). For `gpt-4.1-nano` we instead observe largely the same accuracy on MBPP, MBPP+ and HumanEval and a substantial improvement on HumanEval+ (+3.1%).

While our models have been trained on data from the test portion of MBPP, they have not been specifically trained to solve the MBPP task. They have never seen the natural language instructions. In fact, our models are based on the evaluators (Bobs), which have not been fine-tuned for code generation, yet they still manage to improve or maintain their generation performance.

For tasks oriented towards code generation, it may be beneficial to train a separate model combining the final training datasets of both Alice and Bob.

## 4  Conclusions

We presented a method to enhance the code understanding capabilities of Large Language Models by training them in a self-play setting using the **semantic inequivalence game**.

We motivated the design of this approach with theoretical arguments, demonstrating that it can cover broad domains of real-world programming by being **grounded** in a dataset of examples, while simultaneously having **no theoretical performance cap**. This allows, in principle, for unbounded performance improvements, constrained only by the available computing resources and the learning capacity of the underlying LLMs.

We evaluated our method on a variety of code reasoning tasks, including the challenging **Python builtin identifier swap** benchmark and two security vulnerability detection benchmarks. These evaluations show that our approach learns skills that generalize across tasks and programming languages.

We believe that our method makes a significant contribution to techniques for training LLMs on complex reasoning tasks.

## 5  Limitations and Future Work

Our method has the following limitations, primarily due to our limited budget:

- We primarily fine-tuned `gpt-4o-mini` and `gpt-4.1-nano`, which, while performant, are not a state-of-the-art model. Given more resources, it would be beneficial to repeat the experiments on several more powerful models, including inference-time scaling reasoning models.

- We primarily applied supervised fine-tuning on the OpenAI platform, which likely relies on LoRA-style adaptors instead of full-parameter tuning. It would be valuable to explore reinforcement learning and full-parameter tuning as alternatives. We were able to run LoRA finetuning experiments on the small `Qwen3-4B-Thinking-2507`, but the results were inconsistent.

- It would be beneficial to perform multiple training rounds for Bob, with Alice being trained to convergence between each round for Bob. This could help the models learn powerful code reasoning skills, similar to how AlphaZero learns strong reasoning abilities in Go or Chess through many rounds of self-play.

### Reproducibility Statement

We will release all the code necessary to reproduce our experiments, along with the synthetic training data we generated, upon publication. Exact replication, limited by sampling randomness, should be possible with a modest budget (approximately $600), as long as `gpt-4o-mini-2024-07-18` and `gpt-4.1-nano-2025-04-14` remain available on the OpenAI platform.

### Author Contributions

Omitted to preserve anonymity. To be included in the camera-ready version of the paper.

## Acknowledgments

Omitted to preserve anonymity. To be included in the camera-ready version of the paper.

## Broader Impact Statement

Our proposed method involves training LLMs on synthetically generated data based on existing open-source programming code datasets. We also evaluate our method on open-source datasets. No human experiments were conducted, and therefore, the risk that our experiments have caused harm to individuals or infringed upon anyone's intellectual property rights is negligible.

Our work aims to enhance LLMs' ability to reason about programming code. There is a potential risk that such capabilities could be used for unethical activities, such as hacking computer systems. However, these capabilities can also be used to strengthen computing systems by detecting security vulnerabilities in codebases. We believe that the net societal impact of our research will be positive.

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

## A    Non-decidability of semantic inequivalence

In a semantic equivalence game where the task of the evaluator ("Bob") is to determine whether two programs $P$ and $Q$ are equivalent, there is clear undecidablity due to a trivial consequence of Rice's theorem (Rice, 1953). Since a perfect Bob cannot exist, this results in a "full-employment theorem" (Appel & Ginsburg, 1998) for Alice: in principle, she can always find new ways to fool Bob. This iterative process leads to increasingly stronger Bobs, who in turn train progressively stronger Alices.

However, in the semantic inequivalence game (section 2.1), the programs $P$ and $Q$ given to Bob are guaranteed to be not equivalent, and Bob's task is to find a diverging input $\hat{x}$ such that $P(\hat{x}) \neq Q(\hat{x})$, which is guaranteed to exist. It may be asked whether this constraint on the programs makes the problem any logically easier. We show here that this is not the case.

**Definitions**

Given an arbitrary, but fixed, admissible numbering (programming language) of partial computable functions, we define a program $P$ as an index in such numbering.
With a slight overload of notation, we denote $P(x)$ as the result of evaluating on input $x$ the partial computable function defined by program $P$. Without loss of generality, we consider the inputs of our programs to be the natural numbers and the outputs to be natural numbers plus the special value $\perp$ that denotes non-termination.

The mapping between programs and functions is surjective but not injective: each function can be defined by infinitely many programs. We define two programs $P$ and $Q$ **equivalent** if they define the same function, conversely we define them **inequivalent** if they define different functions, that is, if there exist at least one **diverging input** $\hat{x}$ such that $P(\hat{x}) \neq Q(\hat{x})$.

Given a program $A$ and a natural number $n$, the **halting problem**, denoted by the predicate $Halt(A, n)$, consists of determining whether $A(n) \neq \perp$, which is notoriously undecidable in the general case.

**Theorem A.1.** *There is no perfect evaluator program $\overset{*}{Bob}$ such that, for any inequivalent programs $P$ and $Q$ it computes a diverging input for $P$ and $Q$.*

*Proof.* If programs $P$ and $Q$ have a diverging input for which they both halt producing distinct output values: $P(\hat{x}) = y_p \in \mathbb{N}$, $Q(\hat{x}) = y_q \in \mathbb{N}$ and $y_p \neq y_q$, then $\overset{*}{Bob}$ can compute $\hat{x}$ by dovetailing. The interesting case is when for each diverging input only one between $P$ and $Q$ halts. We show that such diverging inputs cannot be computed in the general case by a reduction to the halting problem.

Given a program $A$ and a natural number $n$, it is possible to algorithmically construct two programs $P^*_{A,n}$ and $Q^*_{A,n}$ defined as follows:

Listing 1: Definition of $P^*_{A,n}$ and $Q^*_{A,n}$

```
def P_A_n_star(x):
    if x == 0:
        A(n)
        return 1
    else:
        while True:
            pass

def Q_A_n_star(x):
    if x == 0:
        return 1
    else:
        P_A_n_star(x-1)
```

By construction, if $A$ halts on input $n$, then $P^*_{A,n}$ and $Q^*_{A,n}$ diverge only on input 1:
$A(n) \neq \perp \iff P^*_{A,n}(1) = \perp \wedge Q^*_{A,n}(1) = 1$,
otherwise if $A$ does not halt on $n$, then $P^*_{A,n}$ and $Q^*_{A,n}$ diverge only on input 0:
$A(n) = \perp \iff P^*_{A,n}(0) = \perp \wedge Q^*_{A,n}(0) = 1$.
They are always equivalent on any other input. Therefore:

Listing 2: Halting decider

```
def Halt(A, n):
    def P_A_n_star(x):  ...  # defined as in Listing 1
    def Q_A_n_star(x):  ...  # defined as in Listing 1
    return Bob_star(P_A_n_star, Q_A_n_star) == 1
```

Since a general program that decides the halting problem cannot exist, then a perfect evaluator for the semantic inequivalence problem $\overset{*}{Bob}$ cannot exist. □

It can be noted that the proof of Theorem A.1 applies in the general case but deviates from the constraints of the semantic inequivalence game in two important aspects:

1. The proof involves distinguishing between the halting behaviour of programs under arbitrary runtime, while in the game programs are checked against the diverging inputs produced by Alice and Bob under a time limit, after which they are assumed to return a special "TIMEOUT" value.

2. In the proof we allow both Bob's input programs $P$ and $Q$ to take a special form that depends on the program $A$ whose halting behaviour is under consideration, while in the game the program $P$ is sampled from a dataset and Alice only controls program $Q$.

It may be asked whether these constraints make Bob's task substantially easier, allowing for a perfect $\overset{*}{Bob}$ to exist, which would imply a performance cap. We show that this is not the case.

In order to address the first point, we note that while the halting problem under a time limit is decidable if we only require the halting detector program to eventually halt, it is still undecidable if the halting detector program has to halt itself within the same time limit of the program it checks[7]. Therefore, by constraining Bob's resource usage, allowing Alice to always have more resources than Bob, and gradually increasing the time limit of the programs, it is possible for Alice to always generate harder and harder instances. Once Bob stops learning, the resource limits can be increased, enabling further learning, in principle forever. In practice, Alice and Bob are implemented as agents based on LLMs operating in chain-of-thought mode, thus resource limits can be enforced by controlling the number of reasoning tokens, or in the long term by controlling the parameter count, layer count, or expert count of the base LLMs[8].

As for the second point, we show that for any **non-trivial program** $P$, Alice can generate a program $\overline{Q_{P,A,n}}$ which checks whether program $A$ halts on input $n$, where by "non-trivial" we mean that there exist at least two distinct inputs $x_0$ and $x_1$ such that $P$ halts on both, returning two distinct values, respectively $y_0$ and $y_1$:

Listing 3: Definition of $\overline{Q_{P,A,n}}$

```
def Q_P_A_n_bar(x):
    if (x == x_0) or (x == x_1):
        if Halt(A, n):  # defined as in Listing 2
            return y_0
        else:
            return y_1
    else:
        return P(x)
```

This is a self-referential construction, where Bob is tasked to analyse a program that invokes Bob itself, thus Bob has to analyse its own behaviour. If Bob was indeed the perfect $\overset{*}{Bob}$, then $P$ and $\overline{Q_{P,A,n}}$ would return different values only on input $x_0$ if $A$ halts on $n$, or only on $x_1$ if it does not, thus solving the halting problem. Note that this construction is still a valid output for Alice even when Bob is not perfect, since

---

[7]This is provable with an argument about program length.
[8]Assuming that LLMs always become better at learning when increasing their resource limits.

$\overline{Q_{P,A,n}}$ will still differ from $P$ on $x_0$ or $x_1$ (possibly on both if the inner call to Bob does not halt), which means that in principle Alice can generate hard examples for Bob from arbitrary source programs, as long as they meet minimal "non-triviality" conditions. In practice, we want the generated programs to run quickly on the CPU without invoking LLMs, so this self-referential construction is unwieldy, but it serves as a proof of concept which shows that arbitrarily complex logic can be added by Alice in the programs it generates, even starting from minimally complex source programs.

## B  Setting a target difficulty

In the implementation of the semantic inequivalence game which we use in our experiment, we instruct the generator "Alice" to create challenge instances for the evaluator "Bob" with a specific target difficulty, defined as 10 times the probability that Bob fails to solve the instance when invoked in sampling mode. Setting the target difficulty always at the maximum value of 10 makes the game equivalent to its original formulation in section 2.1, which, if Alice never produces invalid instances, is a **zero-sum game**.

It may be asked whether this maximally adversarial setting is always ideal. Consider the following Python program that Alice may potentially generate:

Listing 4: Cryptographically hard $Q$ generated for a given $P$

```python
import hashlib

def Q(x):
    try:
        e = str(x).encode("utf-8")
        h = hashlib.sha3_256(e).hexdigest()
        if h == "af9ac3dac56b02f1ea017e7657a9bb7e1778274e31509f134f023e41a5953866":
            return "Bananas"
    except:
        pass
    return P(x)
```

For inputs $x$ that have the specific SHA-3-256 value defined in the code, $Q$ returns the string "Bananas", otherwise it behaves as $P$, therefore, as long as $P$ does not happen to also return "Bananas" for all these specific inputs, they are diverging inputs.

Alice can easily generate this instance by first choosing a diverging input $\hat{x}$ (in this example, the string "correct horse battery staple"), then hashing it and hardcoding its hash value into $Q$, but, in order to solve this instance Bob has to successfully execute a **preimage attack** on SHA-3-256, which is considered a strong cryptographic function (National Institute of Standards and Technology (NIST) & Dworkin, 2015). While this attack is theoretically possible by brute-force search, in practice it would require a runtime longer than the age of the universe, unless perhaps Bob is a cryptanalysis genius and manages to find a serious flaw in SHA-3-256, and even in this case, if the **one-way function conjecture** happens to be true then it is possible to construct asymptotically strong cryptographic hash functions (Levin, 2003), making Bob's task effectively hopeless.

The construction used is our specific example would require Alice to run code in order to compute the hash of its chosen diverging input, which current LLMs are typically not allowed to do in their default configuration and not in our experiments (although some common "LLM agent" setups do allow it), but Alice could still manage to create cryptographic puzzles which are too hard for any practical Bob to solve.

If Alice is instructed to always generate maximally difficult instances, it has an incentive to generate cryptographic puzzles, but since Bob only learns from the instances it can actually solve, this would effectively cause the learning process to stall. In Appendix A we have proven that learning can continue forever in the limit of infinite computing resources, but in reality computing resources are finite, and cryptographic puzzles could stop the learning process as soon as Alice discovers the trick. Even if it never resorts to cryptographic

puzzles, Alice could just learn faster than Bob, eventually overwhelming Bob with instances that it cannot solve and thus stopping the learning process.

Fortunately, we can avoid this problem completely by setting the target difficult to a value lower than the maximum, e.g. 7, corresponding to the current Bob solving the instances with 30% probability. This changes the nature of the game from **zero-sum** to **positive-sum**, where Alice acts as a teacher that challenges Bob with instances which are hard, but not too hard for its current level. As Bob improves, the difficulty of a given distribution of instances decreases, which in turns causes Alice to learn to recalibrate its difficulty estimation and gradually generate more challenging instances, enabling the training process to continue learning interesting coding logic for as long as the capacity of the underlying LLMs is not exceeded.

In our experiments, due to our limited resources, we could not train Alice to the point that it could seriously challenge Bob, thus we always set the target difficulty to 10, but as a training recipe, we do recommend reducing the target difficulty if at some point Bob starts to fall behind.

## C  Prompt templates

**System prompt for Alice**

```
You are an expert computer scientist. Your task is to take a Python 3.10 program and
write a similar program which is not semantically equivalent, which means that there must
exist at least a diverging input example such that the original program and your program
either produce different outputs or exceptions, or one halts and the other one does not
halt. In addition to a program, you need to produce a diverging input example. Start
by carefully analyzing the original program and think of how an example would propagate
through it from the input to the return value, considering how to modify the program
in order to elicit a different behavior. Make sure that the return values or exceptions
raised by your program are picklable.
The original program and your program will be used in a test to evaluate the skill of an
expert computer scientist who will have to produce a diverging example (not necessarily
the same as yours), so make sure that the difference you introduce are not very easy
to understand. You will be given a difficulty level from 0 (easiest) to 10 (hardest) to
target. E.g. difficulty level 0 means that an expert computer scientist in the bottom
decile or above should be able to find a diverging example, difficulty level 9 means that
only an expert computer scientist in the top decile should be able to find a diverging
example, and difficulty level 10 means that only the top 1\% or less of expert computer
scientists should be able to find a diverging example.
Think step by step before writing your program. Use the following Markdown format, making
sure that the following sections are delimited by level 1 headings, since they will have
to be automatically parsed:
# Analysis
step by step analysis. This section can include sub-headings and code blocks
# Generated program
your program inside a Python code block. Do not change the name or signature of the entry
point function
# Diverging input example
your diverging input example as a Python dictionary inside a Python code block
For instance, if the entry point function takes two parameters a and b and your diverging
example is a="foo" and b=42, write:
```python
{
  "a": "foo",
  "b": 42
}
```
```

do not write the expected outputs

**User message for Alice**   As a Python f-string:

```
f"""Difficulty level: {difficulty_level}
Entry point function: {function_name}

```python
{code}
```"""
```

During inference `difficulty_level` is the target difficulty (always 10), during SFT training, for Alice's main examples it is the measured difficulty approximated to the nearest integer, for Alice's difficulty prediction examples it is the string `"Any"`.

**Second user message for Alice**   Used only for the difficulty prediction training examples.

```
Predict the difficulty level of the instance. Just write "Difficulty level: D" where D is
your prediction, do not write anything else.
```

**Second assistant message for Alice**   Used only for the difficulty prediction training examples. As a Python f-string:

```
f"""Difficulty level: {difficulty_level}"""
```

where `difficulty_level` is the measured difficulty.

**System prompt for Bob**

```
You are an expert computer scientist. Your task is to take two Python 3.10 programs and
determine whether or not they are semantically equivalent. Two programs are semantically
equivalent if there exists no diverging input examples such that the original program
and your program either produce different outputs or exceptions, or one halts and the
other one does not halt. If you determine that the two programs are not semantically
equivalent, you also need to produce a diverging input example. Start by carefully
analyzing the two programs and think of how an example would propagate through them from
the input to the return value, considering whether it could elicit a different behaviors.
Think step by step before writing your program. Use the following Markdown format, making
sure that the following sections are delimited by level 1 headings, since they will have
to be automatically parsed:
# Analysis
step by step analysis. This section can include sub-headings and code blocks
# Equivalent?
Yes or No
# Diverging input example
your diverging input example as a Python dictionary inside a Python code block, or
nothing if the two programs are equivalent.
For instance, if the entry point function takes two parameters a and b and your diverging
example is a="foo" and b=42, write:
```python
{
  "a": "foo",
  "b": 42
}
```
do not write the expected outputs
```

Note that we ask Bob to determine whether the two programs are equivalent, even though they never are. This is not strictly necessary, but it potentially makes the task slightly more difficult for Bob, which is beneficial since Bob tends to be much stronger than Alice.

**User message for Bob**   As a Python f-string:

```
f"""Entry point function: {function_name}

Program 1:
```python
{code_P}
```

Program 2:
```python
{code_Q}
```"""
```

The evaluation prompts will be included in the code released upon publication.

## D   Additional Python builtin identifier swap results

**Main results**   Same as Table 2, presented as a bar chart in Figure 5.

**Results on Reasoning Models**   Large Reasoning Models (LRMs) are LLMs which have been specifically trained to solve reasoning tasks, primarily in the domains of math and coding, using Chain-of-Thought reasoning. These models, such as OpenAI `o1` and `o3` and DeepSeek-r1 (DeepSeek-AI et al., 2025) typically generate a large amount of reasoning tokens during inference, hence they are said to perform **inference-time scaling** by trading off speed and cost for quality. In practice, they are very strong but also very expensive.

Our approach could be broadly considered a type of LRM, since it is trained to solve reasoning problems using CoT, though in practice we use a much smaller base LLM and we do not invest nearly as many resources neither during training time nor during inference time.

We evaluate the OpenAI LRMs `o1-2024-12-17` and `o3-mini-2025-01-31` and DeepSeek LRMs `r1` and its distilled version based on Meta `Llama-3.3-70B-Instruct` on the Python builtin identifier swap benchmark. Due to the high cost and low speed of inference, for all these models except `o3-mini-2025-01-31` we only evaluate 10% of the test set. For the OpenAI models we evaluate both on the default prompt and the CoT-style prompt suggested for DeepSeek-r1. We report the results in Figure 6.

The LRMs are much stronger than `gpt-4o-mini`, `gpt-4.1-nano` and our approach, with the full DeepSeek-r1 reaching 94.0% accuracy, which is expected given their training and inference costs.

Given sufficient resources, it would be beneficial as a future experiment to use one of these models as the base model for our approach. We expect that our approach would be complementary to the synthetic data generation techniques used to train these models, resulting in further improvements.

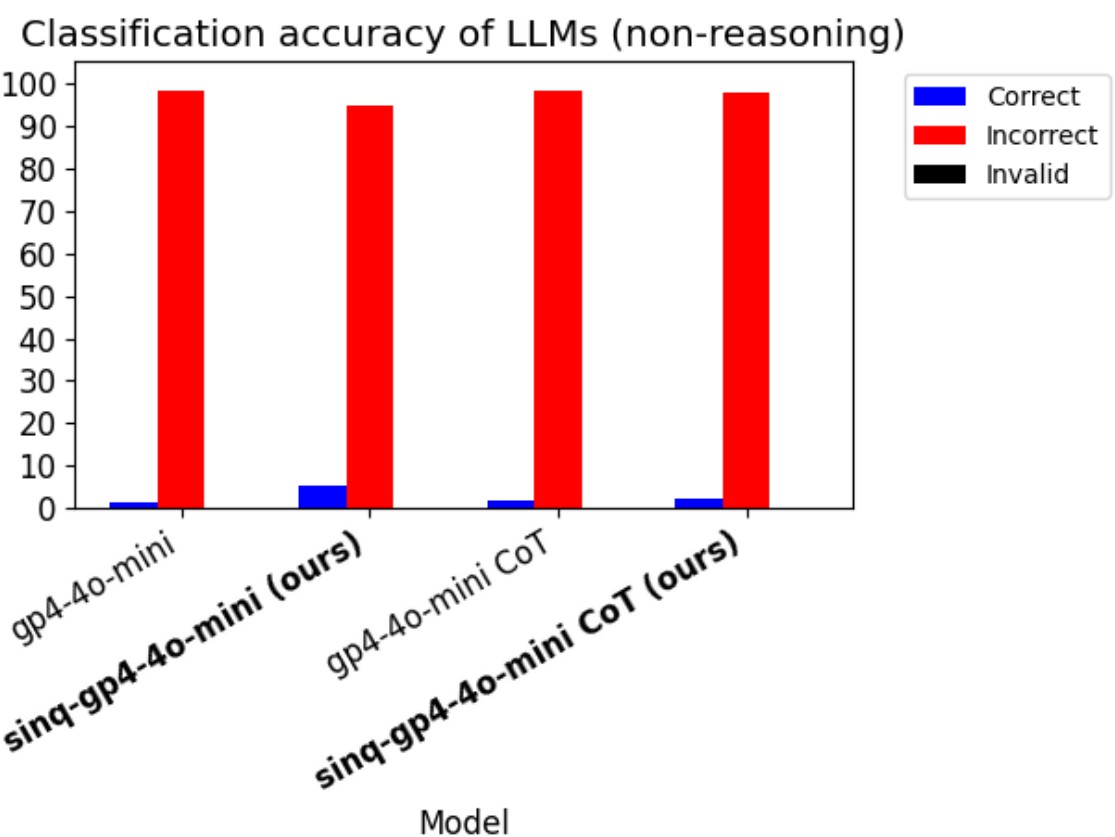

Figure 5: Python builtin identifier swap results for the baseline `gpt-4o-mini` (untrained Bob) and our model `sinq-gpt-4o-mini` (trained Bob), with or without chain-of-thought.

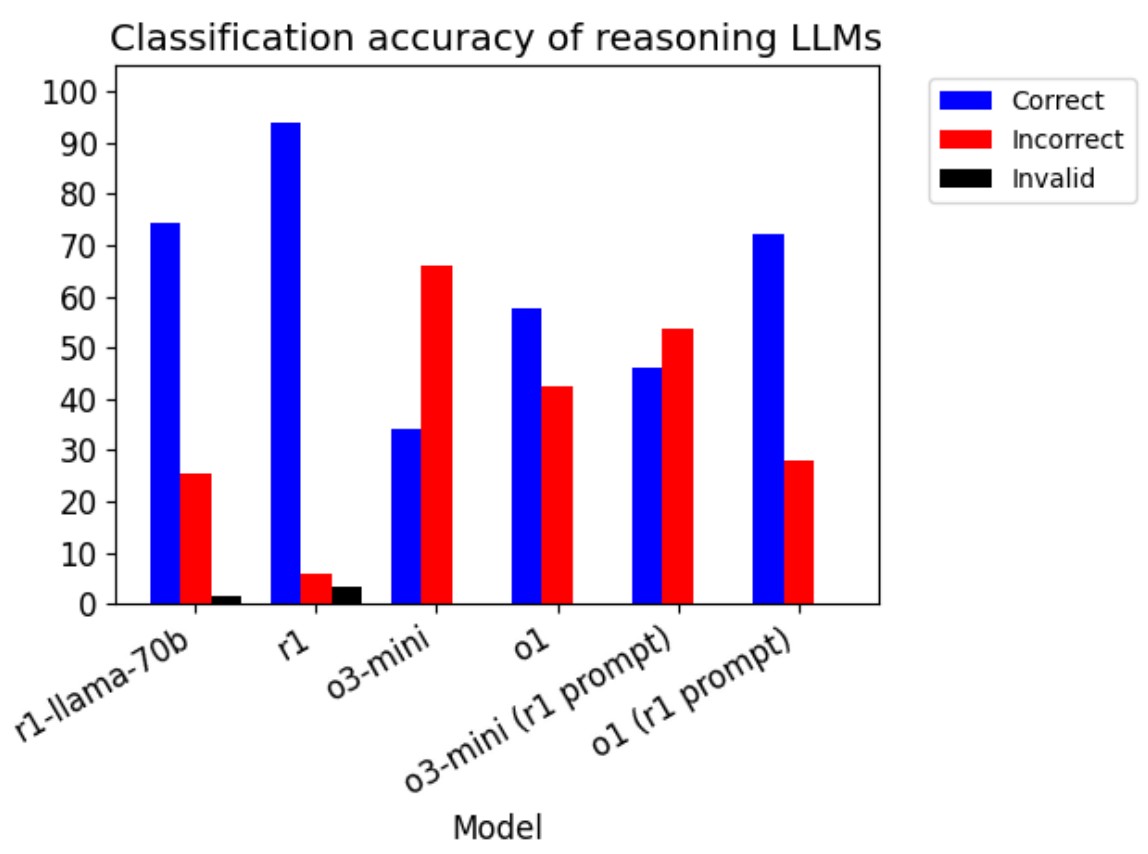

Figure 6: Python builtin identifier swap results for LRMs.

