# OpenReview forum: "Program Semantic Inequivalence Game with Large Language Models"
_TMLR — Rejected by TMLR_

### Review · Reviewer_HKZz · 2025-08-31

**Summary Of Contributions:**

Code-comprehension and generation is a compelling use-case for large language models. In an ideal scenario, an LLM agent could assist in "the detection of preexisting vulnerabilities and adversarial backdoors." To pursue that goal, the authors of this submission propose the following. First a program P is sampled from a dataset. Then a generator and an evaluator  are trained via repeated play of a Semantic Inequivalence (SINQ) Game. In each playthrough, the generator's objective is to produce Q and x such that $Q(x) \neq P(x)$ while the evaluator's objective (given only P and Q) is to identify y such that $Q(y) \neq P(x)$. Ideally, the two parties would be alternatingly trained until convergence via reinforcement learning.

Real-world constraints limit the authors' training (on the MBPP dataset). First, the OpenAI API at the time of their work only allowed supervised fine tuning, so the authors rely on an empirical notion of "difficulty" of P,Q pairs. Second, they had to limit the number of rounds of generator fine-tuning to 7 and the number of rounds of evaluator fine-tuning to 1.

The authors play their evaluator against standard benchmarks to see how well its abilities generalize beyond the SINQ game. For the Python Builtin Identifier Swap benchmark, their work improved the success rate of gpt-4o-mini from 1.65% to 5.35%. For security vulnerability detection (PySecDB / CodeXGLUE Defect Detection), their work improved majority-of-9-voting classification from 82.48% / 55.12% to 82.81% / 56.04%.

*Strengths*
- The adversarial setup is a natural application of a well-established framework to the problem.
- The explicit inclusion of limitations & constraints is appreciated (Section 3.1, Section 5), as is the use of concrete examples (Example 1, Section 3.3)
- The specification of SINQ is rigorous and precise.
- The writing is easy to follow overall

*Weaknesses*: Some content/context is missing
1. Accuracy tests are reported with raw percentages, but not standard deviation thereof; are we confident that the degree of improvement is behavior that a customer can consistently expect? As an example, the SINQ fine-tuning could be the reason for the small improvement on the PySecDB benchmark, but perhaps it is due to fluctuations in the LLM
2. What was the motivation for running "main set of experiments on OpenAI gpt-4o-mini-2024-07-18 as the base LLM for both
Alice and Bob"? Why that iteration in particular? And why not other LLMs?
3. How much of "MBPP Train" and "MBPP Test" were consumed to generate Table 1? Was it everything, or a small fraction? The same question applies to Table 3.

**Audience:**

Yes

**Audience Explanation:**

Code analysis and generation is an attractive area of LLM research, so I imagine the community will appreciate both the experiments and the new SINQ game

**Claims And Evidence:**

No

**Claims Explanation:**

I would have appreciated more information regarding W1 and W2. If there are organizational or financial constraints that forced one version of GPT to be used a small number of times, that is understandable, but this should be reflected in a revision.

**Requested Changes:**

See explanation for the "No" above. Also,
- Why use Alice/Bob and not generator/evaluator?
- Why include the factor of 10 in the definition of difficulty?
- Were there any attempts to automatically generate diverging source code prior to the advent of LLMs? (i.e. although the SINQ game is new, is there precedent for what "Alice" does?) Regardless of what the answer to that question is, it should be included

---

> ### Author Response · Authors · 2025-11-15
> **Update**
>
> We have updated our paper with results on OpenAI gpt-4.1-nano , which were successful and confirmed our previous results on gpt-4o-mini, and an attempted training run on Qwen3-4B-Thinking (the smallest version available on Hugging Face) fine-tuned with LoRA, which was unsuccessful possibly due to small model or adaptor size. We lack the resources to fine-tune larger models. Nevertheless, we believe that our results confirm that our approach works on models of sufficient strength.

---

### Review · Reviewer_G45b · 2025-09-01

**Summary Of Contributions:**

This paper focuses on synthesizing code reasoning training data based on a semantic inequivalency game, termed SInQ, with large language models (LLMs). The proposed game consists of a generator agent that creates semantically distinct program variants, while an evaluator agent identifies inputs that induce different behaviors. Such a kind of adversarially paradigm improves models ability to understand the reasoning patterns of code, and is empirically demostrated with tasks like cross-language vulnerability detection, and the challenging Python builtin identifier swap benchmarks.

**Audience:**

Yes

**Audience Explanation:**

The work addresses an important topic in LLM research, which is of interest to TMLR’s readership:
1. The introduction of a semantic inequivalence game to generate synthetic training data is novel, and will interest readers studying self-play, program synthesis, and LLM reasoning.
2. The fact that training on synthetic Python code data yielded improvements on diverse tasks (including C/C++ vulnerability detection and identifier swap) will attract readers interested in related scenarios of transfer learning with LLMs.

**Claims And Evidence:**

No

**Claims Explanation:**

While the paper presents interesting ideas and some supporting results, the evidence falls short in the following aspects:
1. The experiments were only conducted on one small-scale model due to resource constraints, with no exploration on larger, more diversified LLMs. This makes it difficult to assess whether the claimed improvements generalize broadly or are specific to the chosen small-scale model.
2. While there are gains on some benchmarks (e.g., MBPP, vulnerability detection, identifier swap), others show marginal or even negative changes (e.g., a slight drop on HumanEval+). The improvements are small and not always statistically analyzed; it would be better if there were some additional evidence to strongly support the broader claims of “significant” or “generalizable” advances.
3. The generator agent (“Alice”) was not trained to converge, and the evaluator (“Bob”) was trained for only a single round. I wonder about the rationality of the design of the training paradigm, and the clear motivation or explanation for such proposal is needed.

**Requested Changes:**

This paper presents an interesting framework to synthesize the training data for improving LLM reasoning, while the presentation can be improved by considering the following suggestions:
1. Expand experiments to include larger and more diverse base models (e.g., stronger reasoning LLMs or inference-time scaling models). This would validate whether the method scales and generalizes beyond a single mid-sized baseline.
2. Provide multiple rounds of self-play training where Alice is trained to converge before updating Bob. And provide some explanation about the rationality of the training design.
3. Add statistical significance testing and ablations (e.g., varying difficulty targets, evaluating against cryptographic-puzzle-like cases) to strengthen the empirical claims.

---

### Review · Reviewer_vHYx · 2025-10-14

**Summary Of Contributions:**

This paper proposes a novel self-play framework for code reasoning, introducing the Semantic Inequivalence Game (SInQ), in which two LLM agents the generator and the evaluator,—train each other by generating and solving pairs of semantically distinct programs. The key idea is to create synthetic code reasoning data without relying on human annotation or explicit natural language supervision. generator generates semantically inequivalent program variants and corresponding diverging inputs, while evaluator learns to detect inequivalence by producing counterexamples.
This paper implement the approach using GPT-4o-mini and evaluate the trained “Bob” model (sinq-gpt-4o-mini) on multiple benchmarks. Results show consistent improvements in program semantics reasoning—notably in Python Builtin Identifier Swap, vulnerability detection, and even cross-language generalization to C/C++ vulnerability detection tasks.

Strengths

1. The paper addresses a meaningful and important problem.
2. The idea is novel, formulating program reasoning as a self-play “semantic inequivalence” game.

Weaknesses

1. The experiments are conducted only on GPT-4o, leaving the generalizability of the method uncertain. Could you evaluate your approach on other LLMs, particularly open-source ones such as DeepSeek or Qwen?

**Audience:**

Yes

**Audience Explanation:**

The study is meaningful, and the proposed method is novel.

**Claims And Evidence:**

No

**Claims Explanation:**

Although the paper conducts experiments on several code datasets, the evaluation is limited to GPT-4o, making its generalization ability unclear.

**Requested Changes:**

The experiments are conducted only on GPT-4o, leaving the generalizability of the method uncertain. Could you evaluate your approach on other LLMs, particularly open-source ones such as DeepSeek or Qwen?

---

> ### Author Response · Authors · 2025-10-14
>
> Thanks for your review. We'll attempt additional experiments and post a revision.

---

> ### Author Response · Authors · 2025-11-15
> **Update**
>
> We have updated our paper with results on OpenAI gpt-4.1-nano , which were successful and confirmed our previous results on gpt-4o-mini, and an attempted training run on Qwen3-4B-Thinking (the smallest version available on Hugging Face) fine-tuned with LoRA, which was unsuccessful possibly due to small model or adaptor size. We lack the resources to fine-tune larger models. Nevertheless, we believe that our results confirm that our approach works on models of sufficient strength.

---

### Review · Reviewer_Uzq7 · 2025-12-06

**Summary Of Contributions:**

The paper proposes a self-play training framework—SInQ—where a generator LLM (Alice) produces program variants that are semantically inequivalent to a source program, and an evaluator LLM (Bob) must find diverging inputs. This yields synthetic training data emphasizing semantic reasoning, not surface patterns. The authors fine-tune small OpenAI models using this game and report improvements on several code-reasoning benchmarks, including Python Identifier Swap and vulnerability detection.

The paper presents a novel, well-motivated self-play framework that teaches LLMs genuine semantic program reasoning, with clear theoretical grounding and consistent improvements across multiple code-reasoning tasks. However, evaluation is limited—no strong baselines, only one self-play round, and reliance on MBPP—so broader scalability and the core claim of open-ended improvement remain unverified.

**Audience:**

Yes

**Audience Explanation:**

The paper introduces a novel, theoretically grounded self-play framework for enhancing LLM semantic reasoning—an area crucial to the development of reliable code models and AI safety. The method provides a new way to generate rich supervision without human annotations, and the observed transfer to real-world tasks (e.g., vulnerability detection) suggests practical impact for both machine learning researchers and programming languages practitioners.

**Broader Impact Concerns:**

The main broader-impact concern is that strengthening LLMs’ ability to reason about subtle code semantics may also enhance their capacity to analyze or manipulate software in ways that could be misused—for example, to identify vulnerabilities or craft exploit-triggering inputs. While the paper frames this capability as beneficial for security auditing, the dual-use nature of improved program-analysis skills warrants explicit discussion, including potential misuse scenarios and mitigation strategies. Additionally, because the method trains models to generate semantically inequivalent code variants, there is a small but real risk that such techniques could be adapted to automate the creation of harmful or deceptive program transformations unless appropriate safeguards and usage policies are in place.

**Claims And Evidence:**

No

**Claims Explanation:**

- **Limited baselines.** The paper only compares SInQ-trained models to their own base models. It does not compare against alternative synthetic-data generation approaches (e.g., mutation-based data, equivalence datasets, test-generation self-play), making it difficult to assess relative benefit.

- **Weak evidence for open-ended self-play improvement.** Although the paper claims unbounded potential improvement, the experiments include only one training round. No multi-round Alice↔Bob co-evolution is demonstrated.

- **Difficulty metric instability.** Difficulty is estimated from 10 samples at a temperature of 1.0, which may be noisy. The Qwen experiments show instability consistent with this issue.

- **Narrow source distribution.** Alice is always trained on MBPP programs, which are short, single-function Python snippets. This limits diversity and may explain why downstream gains remain modest in absolute terms.

**Requested Changes:**

1. Include stronger baselines – Compare SInQ against alternative synthetic data or mutation-based code reasoning approaches to demonstrate the method’s distinct advantage.

2. Demonstrate at least one full Alice–Bob iteration cycle – Show that self-play can improve beyond a single Bob training round, supporting the claim of open-ended learning.

3. Broaden the training source beyond MBPP – Incorporate more diverse program types to validate that the method generalizes beyond short Python snippets.

---

### Author Response · Authors · 2025-10-27
**Response**

We have run additional experiments on gpt-4.1 and qwen-3 models, we will update the paper with the results in the following couple days.

---

> ### Author Response · Authors · 2025-11-15
> **Update**
>
> Updated with Qwen3-4B-Thinking results.

---

### Decision · Action_Editor_xVAe · 2025-12-05

**Recommendation:** Reject

**Additional Comments:**

The submission presents promising idea. However, all the reviewers are in concensus that the current empirical validation is too narrow.  A major revision with broader experimental coverage, clearer justification of training dynamics, ablations, variance estimates, and stronger empirical results would make it a very strong contribution. The authors are strongly encouraged to resubmit once the technical issues and evidentiary gaps have been addressed.

**Audience:**

Yes

**Audience Explanation:**

The topic is timely and relevant for researchers working on code reasoning with LLMs, self-play or adversarial data-generation frameworks.

**Claims And Evidence:**

No

**Claims Explanation:**

The paper explores a self-play–style Semantic Inequivalence Game (SInQ) framework to synthetically generate program-reasoning training data using LLM agents (“generator” and “evaluator”). Reviewers consistently found the idea promising; however, all three reviewers raised concerns about claims being too broad and question that evidence fully supports the strength and breadth of the paper’s claims. In particular, results rely almost exclusively on GPT-4o-mini with only a small number of follow-up experiments on gpt-4.1-nano and a failed attempt on Qwen3-4B-Thinking. Also many training details and hyper-parameters are missing.

**Resubmission Of Major Revision:**

The authors may consider submitting a major revision at a later time.